# Chitin is a functional component of the larval adhesive of barnacles

Nick Aldred [1], Vera Bin San Chan [2,3], Kaveh Emami[4], Keiju Okano[5], Anthony S. Clare[1] & Andrew S. Mount[2]*

Barnacles are the only sessile crustaceans, and their larva, the cyprid, is supremely adapted for attachment to surfaces. Barnacles have a universal requirement for strong adhesion at the point of larval attachment. Selective pressure on the cyprid adhesive has been intense and led to evolution of a tenacious and versatile natural glue. Here we provide evidence that carbohydrate polymers in the form of chitin provide stability to the cyprid adhesive of *Balanus amphitrite*. Chitin was identified surrounding lipid-rich vesicles in the cyprid cement glands. The functional role of chitin was demonstrated via removal of freshly attached cyprids from surfaces using a chitinase. Proteomic analysis identified a single cement gland-specific protein via its association with chitin and localized this protein to the same vesicles. The role of chitin in cyprid adhesion raises intriguing questions about the evolution of barnacle adhesion, as well as providing a new target for antifouling technologies.

[1] School of Natural and Environmental Sciences, Newcastle University, Newcastle Upon Tyne NE1 7RU, UK. [2] Biological Sciences, Clemson University, Clemson, SC 29634, USA. [3] Ifremer, Physiologie Fonctionnelle des Organismes Marins UMR 6539 LEMAR (CNRS/UBO/IRD/Ifremer), CS 10070, F-29280, Plouzane, France. [4] Biosciences Institute, Newcastle University, Newcastle Upon Tyne NE1 7RU, UK. [5] Department of Biotechnology, Akita Prefectural University, Akita, Japan. *email: mount@clemson.edu

Barnacle cyprids have been described as the pinnacle of sessile evolution[1]. While cirripedes can differ markedly as adults, from the familiar acorn barnacles to more unusual parasitic forms, they all share conserved morphology at the cyprid stage[2]. Cyprids have numerous specialized adaptations for sensing and attaching to appropriate surfaces, such as compound eyes that appear only briefly in the cyprid before being lost during metamorphosis to the adult[1]. A pair of highly dextrous antennules enable bipedal walking during the pre-attachment exploratory phase[3–5] and 11 individually identifiable sensory setae[6] set cyprids apart from the settling larvae of other marine invertebrates. Uniquely among Crustacea, many barnacle species must position themselves close to a future mate at attachment but also allow for growth and adhesion in the adult form[7], which has led to supreme surface selectivity. The success of barnacles, with around 1445 extant species[8], is not only dependent on their ability to choose a surface well, but also on effective adhesion to the chosen surface. While selection of an inappropriate settlement site would be an inconvenience possibly mitigated by sperm casting[9] or the proximal settlement of conspecific larvae, failure of adhesion during metamorphosis to the adult form would be fatal. Despite its pivotal role in the lifecycle of barnacles, the composition of cyprid permanent adhesive (syn. cement) has been evaluated only superficially, and the molecular mechanisms underpinning adhesion are poorly understood.

The historical barriers to the characterization of cyprid adhesive have been largely technical. The tiny size of the adhesive plaque (less than 100 μm in diameter) and its availability for only a few hours between cyprid settlement and metamorphosis have made direct collection of the adhesive for analysis practically impossible. Nevertheless, there are significant drivers for understanding barnacle adhesion, foremost among which is the conspicuous role of barnacles in marine biofouling communities and the resurgent interest in all manner of biological adhesion systems as inspiration for surgical glues[10]. Annual carbon emissions from shipping account for ~3% of the global total[11], equivalent to an economy the size of Germany, and it is said that the 15 largest container ships emit more $NO_x$ and $SO_x$ than all of the world's cars combined[12]. Biofouling of ships' hulls contributes significantly towards fuel consumption (up to 86% for heavy barnacle fouling[13]) and thus greenhouse gas emissions, as well as to the translocation of problematic invasive species[14] with associated impacts on local marine biodiversity. Interference with barnacle adhesion is therefore of significant interest to the marine community. Conversely, glues capable of adhesion in wet and contaminated environments have recently emerged based upon the attachment systems of marine mussels[15], slugs[16] and tubeworms[17], and have the potential to revolutionize industries as diverse as human medicine and the production of ply-wood.

The cement glands of cyprids (Fig. 1b) are epithelial in origin and contain two cell types; so-called α- and β-cells[18–20]. The α-cells contain small, spherical, protein-rich secretory vesicles. The β-cells contain larger and more irregular vesicles that have less protein and significant lipid content[20]. When secreted to the surface, the resulting adhesive plaque that embeds the antennules has a central protein-rich core surrounded by a lipid-rich outer layer. Aside from the general morphology and identification of a small number of adhesive proteins analogous to those in the adult, e.g.[21], little is known about the chemistry of adhesion or cohesion. While oxidation of catechol groups to quinones and their subsequent crosslinking ('quinone tanning') was historically proposed as a curing mechanism[18], as in mussel byssal threads and arthropod cuticle[22,23], this hypothesis lacks compelling evidence in vivo. Adult barnacle adhesion is linked to the moulting cycle, however e.g.[24,25], and cyprid cement secretion is also quickly followed by ecdysis. It is therefore possible, considering

the evidence provided by Walker[18], that both adult and larval adhesion processes evolved from a modification of the cuticle secretion process. A key piece of evidence to support further exploration of this hypothesis would be the presence of chitin, an essential crosslinker of arthropod cuticular proteins[22,26]. While chitin, a homopolymer of N-acetyl-D-glucosamine, is ubiquitous, in arthropods it may be present in different forms. In insect cuticle, chitin is usually present as nanofibers that are produced via a membrane-bound chitin synthase and crosslinked by hydrogen bonding between proteins with conserved chitin-binding domains (CBDs)[27]. However, stronger bonds between chitin and proteins are also common when chitin is present as a protein glycosylation, such as in Cancer pagarus[28]. The origins of chitin in arthropods were examined by Hackman[29] who described chitin covalently linked to proteins and chitin produced in isolation via chitin synthase. Adopting his terminology for discussion of the present results, we thus refer to homopolymers of N-acetyl-D-glucosamine covalently linked to proteins (glycosylation) as native chitin[29], while nanofibers extruded via a chitin synthase are referred to simply as chitin.

Here we identify chitin in the cement glands and secreted adhesive of barnacle cyprids and identify a cement gland-specific protein by means of its associated chitin. Further, we demonstrate that when the chitin component of the glue is attacked using a chitinase, the adhesive bond is compromised. It is therefore concluded that chitin is a functional component of the larval adhesive of barnacles.

## Results

**Identification of chitin in the cyprid cement**. A strain of Escherichia coli (T7 Express cells, New England Biolabs) transfected with CBD-coding plasmid pYZ205 was used to develop a chitin-specific fluorescent probe, as described in a previous study[30]. Cement glands from 3-day-old Balanus amphitrite cyprids (Fig. 1a, b) were labelled using the CBD conjugated with SNAP-reactive Alexa Fluor 546 (CBD-546, New England Biolabs) and counter-stained with DAPI (Fig. 1c, d, g). A fluorescence signal from CBD-546 was detected within the β-cells of the cement gland (Fig. 1d). The chitin signal was heterogeneous within individual β-cell vesicles (Fig. 1c) and appeared as a clear ring surrounding each vesicle, co-located with the vesicular plasma membrane (Fig. 1e, f). There was no detectable auto-fluorescence from the gland contents at the wavelengths and intensities used for imaging.

The compositional arrangement of lipid (yellow) and protein (green) in freshly secreted adhesive plaques (Fig. 1h, i) was consistent with previous studies, in terms of an outer, lipid-rich layer surrounding a protein-rich core[20]. However, an intermediate chitin-rich layer (red) was also observed. The chitin (Fig. 1i; red) was concentrated between the central proteinaceous core (green) of the cement plaque and the outer lipidic region. Optical sections through plaques stained for protein and chitin (Fig. 1j) confirmed the result, identifying a strong chitin signal surrounding the protein-rich core. The discovery of chitin in the β-cells, and at an intermediate zone in the secreted cement plaque, was reproduced using biotinylated wheat germ agglutinin (WGA) which also identified chitin, however with less specificity than CBD-546. WGA has a number of experimental advantages over CBD-546. First, it is readily available commercially, as opposed to CBD which requires recombinant production in the laboratory. Biotinylated WGA is also directly applicable to immunoprecipitation and standard two-stage western blotting, whereas CBD requires an intermediate anti-SNAP step. This more experimentally flexible chitin-binding moiety was therefore used in subsequent experiments. Histological comparison of the two

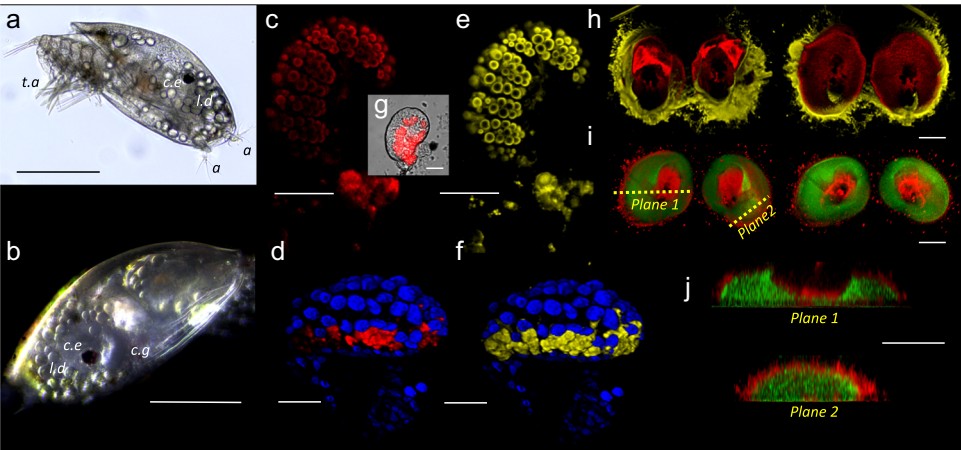

**Fig. 1 Identification of chitin in the cement glands of cyprids. a** A brightfield image of a *B. amphitrite* cyprid, viewed from beneath during surface exploration (t.a. = thoracic appendages used for swimming, a = antennules used for walking, l.d. = lipid droplets for energy storage, c.e. = compound eye). **b** A darkfield image of a cyprid including the cement gland (c.g). **c** Chitin-rich vesicles in the cement gland labelled for chitin (red; CBD-546). **d** Another cement gland labelled for both chitin (red; CBD-546) and nuclei (blue; DAPI). According to **e**, **f**, these vesicles are also rich in lipid/lipid membrane (yellow; CellMask^TM). **g** A brightfield/fluorescence image of a cement gland and the location of the fluorescence in **c**, **e**, **h**. The cement deposited onto a surface embedding two antennules, viewed from above (left) and below (right) shows that chitin (red; CBD-546) is surrounded by lipid (yellow; CellMask^TM). **i** The cement deposited onto a surface embedding two antennules, viewed from above and below shows that protein (green; FITC) is surrounded by chitin (red; WGA). Transverse sections in planes 1 and 2 are presented in **j**. (Scale bars: **a** and **b** = 200 μm, **c**–**i** = 20 μm.).

probes confirmed that signals from CBD-546 (Fig. 1c) and WGA (Fig. 2a, b) identified the same intracellular locations, confirming the specificity of WGA to chitin in this instance. More broadly, the reduced specificity of WGA compared to CBD-546 was evident in, e.g., the labelling of the antennular cuticle by WGA (Fig. 1j), while CBD-546 did not label crosslinked chitin in the cuticle (Fig. 1h). Staining with Nile Red confirmed the lipid-filled lumen of the β-vesicles reported previously[20] (Fig. 2c). When compared to confocal images, super-resolution gSTED imaging indicated reduced thickness of the chitin layer surrounding the β-vesicles, suggesting that the thickness measurement of this layer was probably still diffraction limited and likely less than the apparent 0.08 μm (Fig. 2c, d).

**Chitin is essential for the successful adhesion of cyprids**. To determine if the chitin found in the cyprid cement contributed directly to adhesion, cyprids attached by their permanent adhesive were exposed to a 1 mg/mL solution of chitinase from *Streptomyces griseus*. Control animals were exposed to solutions of heat-inactivated (HI) chitinase or artificial seawater (ASW). Those exposed to chitinase began to detach from the surface of a polystyrene Petri dish after 4 h of exposure. After 8 h, virtually all cyprids had become detached (Fig. 3a; 98% ± 2 (SE)). Cyprids that were exposed to HI and ASW (Fig. 3a) all remained attached. Individuals that metamorphosed prior to chitinase exposure (juvenile barnacles) also remained attached. The mode of chitinase-induced adhesive failure in the cyprid cement was either removal of the embedded antennule(s) from the adhesive deposit (Fig. 3b, c), in which case the antennules themselves remained visibly undamaged, or detachment of the intact adhesive plaque from the surface with the antennule(s) remaining embedded (Fig. 3d, e), again indicating action of chitinase on the adhesive plaque rather than the cuticle of the antennule. All barnacles were alive at the conclusion of the assay.

**Identification of putative chitin–protein complexes**. Chitin is rarely found in a pure state in nature and is most commonly

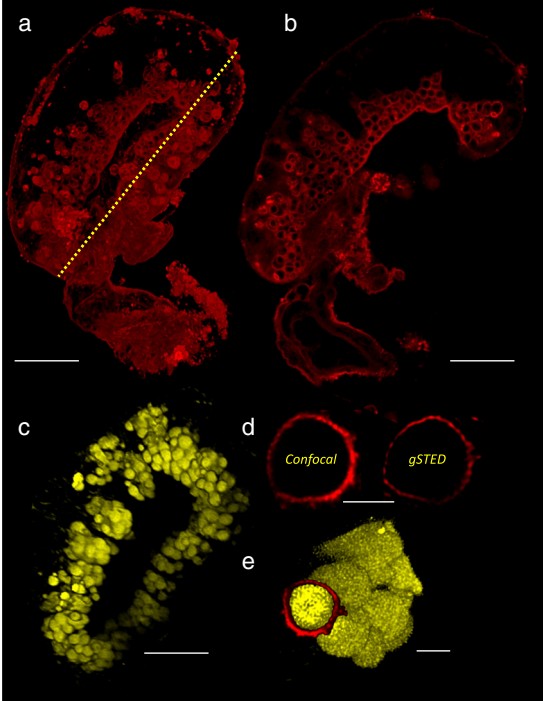

**Fig. 2 Confirming the chitin specificity of wheat germ agglutinin (WGA). a** A cement gland stained using WGA (red). The longitudinal section indicated by the yellow dashed line is presented in **b**, **c**. The lipid-rich intracellular vesicles of the β-cell, stained using Nile Red (yellow). **d** A confocal image and a gSTED image of a WGA-labelled β-cell vesicle (3.7 μm in diameter) shows the thickness of the WGA reactive layer as 0.2 μm and 0.08 μm, respectively. **e** A composite image created using separate, independently acquired images (**c**, **d**) illustrates the lipid-filled lumen and surrounding chitin of a β-cell vesicle in a cement gland cell. (Scale bars: **a**, **b**, **c** = 20 μm, **d** = 2 μm, **e** = 2 μm.).

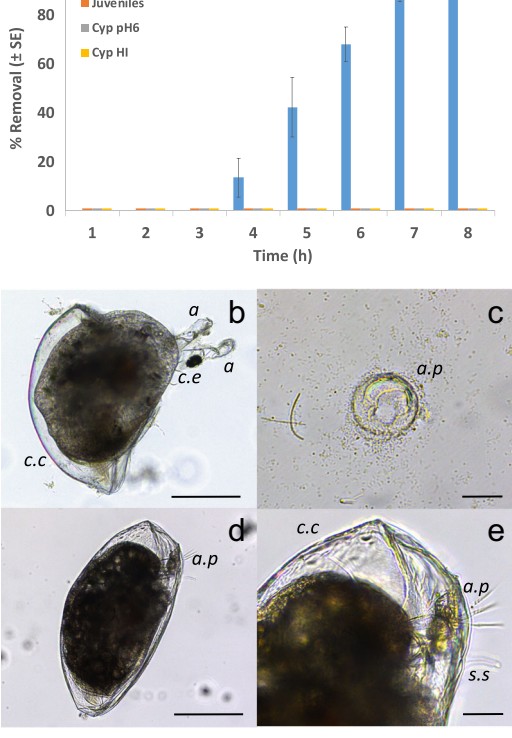

**Fig. 3 Removal of permanently adhered cyprids using chitinase.**
**a** Removal of attached cyprids during exposure to a 1 mg/mL solution of chitinase, compared to controls (Cyprids = settled cyprids exposed to 1 mg/mL chitinase at pH 6.0, Juveniles = settled and metamorphosed individuals exposed to 1 mg/mL chitinase at pH 6.0, Cyp pH6 = settled cyprids in seawater at pH 6.0, Cyp HI = settled cyprids in heat-treated chitinase at pH 6.0). Data were zero for all treatments except Cyprids. Raw data and alternate plots available in Supplementary Data 2 ($n = 6$ biologically independent exposures). **b** A partially metamorphosed individual that became detached from the surface during the process of ecdysis and chitinase treatment. In this case, the intact antennules (**a**) came free from the adhesive deposit (a.p.), which remained on the surface as shown in **c**. **d** A cyprid earlier in the process of metamorphosis was removed from the surface with both antennules still embedded in the adhesive plaque (a.p.) (**e**). (c.c. = cyprid carapace, c.e. = compound eye, both lost during metamorphosis, s.s. = sensory setae. Scale bars: **b** and **d** = 200 μm, **c** and **e** = 50 μm).

complexed with proteins[31]. To investigate the presence of chitin with cement proteins, we isolated cement gland-specific proteins that had affinity to WGA. Samples for gel electrophoresis were produced from 3-day-old cyprids of *B. amphitrite* and included: 100 cement glands dissected from 50 individuals (100 glands), and 50 whole cyprids with their paired cement glands. Subsequent to lysis and homogenization, sodium dodecyl sulphate-polyacrylamide gels (SDS-PAGE) separated the total proteins by mass. Proteins were transferred onto PVDF membrane for western blotting and incubated with biotinylated WGA. Only two protein bands showed specific affinity to WGA, confirming the stringency of the WGA probe. Both were above 120 kDa in mass (Fig. 4). The two bands were clear in both the 100 glands and 50 cyprids lanes (Fig. 4). However, the equivalent bands had different intensities in the whole cyprid extract, with the higher MW band (upper band) being much more intense (Fig. 4). In all cases, western blots without a primary lectin/antibody resulted in blank membranes, excluding nonspecific recognition.

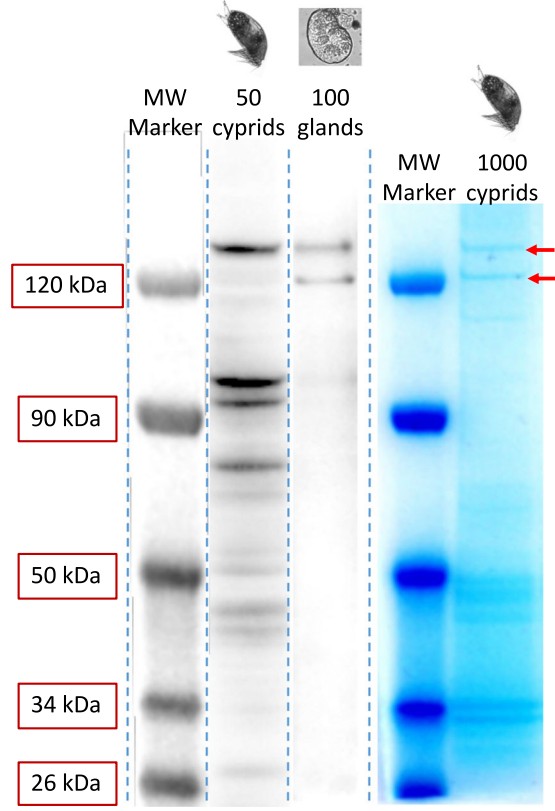

**Fig. 4 Identification of putative chitin–protein complexes in the cyprid adhesive.** Western blot (WB) lanes containing extract from 50 whole cyprids and 100 cement glands, labelled using WGA. Two clear bands were evident in the 100 glands lane. Exposures of WB were optimized for presentation purposes for each lane. Original blots are available as Supplementary Figs. 2 and 3. The molecular weight marker applies to both WB lanes. Lanes stained using colloidal Coomassie blue highlight proteins retained after immunoprecipitation with WGA. The original gel can be found in Supplementary Fig. 4. Red arrows highlight the two protein bands (from 1000 whole cyprids) that were excised for MS/MS characterization.

In order to isolate cement gland proteins with WGA-binding specificity and obtain the quantity required for mass-spectrometry, whole cyprid protein extracts from 1000 individuals were subjected to immunoprecipitation using WGA conjugation, and the precipitate was resolved using SDS-PAGE. Once stained using colloidal Coomassie blue, the lanes containing precipitate of WGA-binding proteins presented the same two bands previously identified in western blots (Fig. 4). The proteins in these bands were isolated and subjected to peptide mass fingerprinting by LC-MS/MS. Peptide mass spectra from these two bands were searched against an all-stages *B. amphitrite* transcriptome[32]. From these searches, numerous peptide groups (putative proteins) were identified in the upper and lower cement gland bands. These peptide groups were down-selected (Supplementary Data 1; see methods) and cross-referenced to a cement gland-specific library generated in parallel for another barnacle species, *Megabalanus rosa*. None of the peptide groups in the top band shared any similarity with proteins in the *M. rosa* cement gland-specific library. On the other hand, four groups (Supplementary Table 1) in the lower band matched a *M. rosa* cement gland-specific protein, Mr-lcp1-122k (NCBI accession number MK490677; see methods, Supplementary Table 1 and Supplementary Fig. 1), with a hypothetical molecular mass of 122 kDa (121670.63 Da, 1082 amino acids, pI = 11.67) and no homology to available protein sequences (NCBI). The four *B. amphitrite*

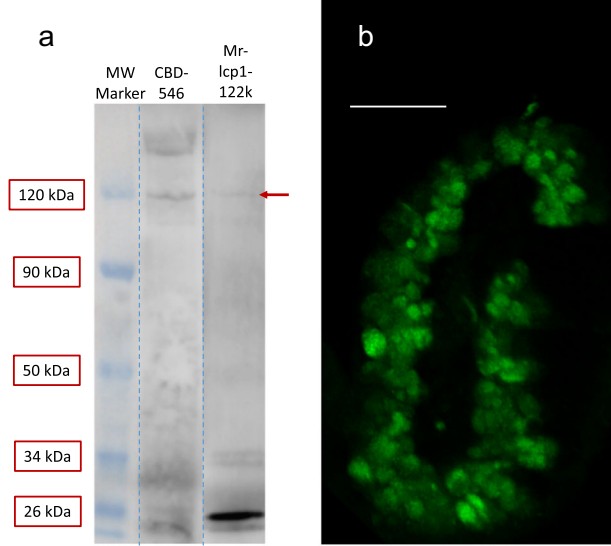

**Fig. 5 Verification of Mr-lcp1-122k homologue location and presence of chitin. a** Western blot (WB) lanes containing whole cyprid extract showed a number of bands that reacted with the CBD and one band that reacted with an antibody for the N-terminal peptide of Mr-lcp1-122k (weak band, red arrow), both of which occurred at ~120 kDa. The molecular weight marker applies to both lanes. Exposures of WB were optimized for presentation purposes and the originals can be found in Supplementary Figs. 5 and 6. **b** A cement gland stained using the N-terminal antibody, illustrating weak fluorescence in the β-vesicles. No fluorescence evident in negative control (scale bar: **b** = 20 μm).

contigs matching Mr-lcp1-122k, taken together, were confirmed to have 85% coverage of the *M. rosa* sequence, between 65 and 72% identity with their *M. rosa* equivalents and between 20 and 59% coverage by the peptides generated in MS/MS experiments (Supplementary Table 1 and Supplementary Fig. 1).

To confirm that Mr-lcp1-122k had been selected via the presence of chitin/native chitin, and not simply by the presence of a small number of N-acetyl-D-glucosamine residues in a more complex glycosylation, sufficient for WGA recognition, the western blotting procedure was repeated using the CBD and with a polyclonal antibody raised to an N-terminal peptide of Mr-lcp1-122k (YVYPRAISHRRPVRYLIQR; Fig. 5a). Significant chitin contamination in whole-cyprid extract caused some difficulties for the three-stage CBD western blot method, however after optimization several clear bands could be recognized, one of which was present at 120 kDa (Fig. 5a) in a similar location to the lower band containing the Mr-lcp1-122k homologue in Fig. 4. The antibody raised to the Mr-lcp1-122k N-terminal peptide also provided a single very faint band at 120 kDa. The antibody was then used to localize the Mr-lcp1-122k homologue in the *B. amphitrite* cyprid cement gland, where it produced weak staining in the vesicles of the β-cells (Fig. 5b).

## Discussion

Although the molecular mechanisms controlling barnacle adhesion are poorly understood, it is widely accepted that the barnacle system differs from other, better-described, marine bioadhesion processes involving L-DOPA, coacervates or both[10]. Proteins with silk-like domains have been identified in adult barnacle adhesive[33], but their precise role is unknown. Although the sequence similarity of known adhesion proteins[34] can be low between barnacle species[35], there is often significant homology in terms of relative amino acid composition, secondary structure and isoelectric points that nevertheless suggest conservation of function.

Despite being the obvious target for fouling-control technologies and having evolved an adhesive on which the survival of the animal depends, the cyprid stage of barnacles has received scant attention. We hypothesized that adhesion in barnacles could have evolved from the process of ecdysis whereby arthropods periodically moult their exoskeleton in order to grow and, in so doing, form new cuticle that hardens over time via the crosslinking of specific proteins with chitin. While robust pursuit of this hypothesis would represent a significant undertaking, the presence of chitin in the cyprid adhesive could provide initial support to the theory. A molecular probe (CBD) was produced to identify chitin polymers specifically. This probe detected only simple chitin homopolymers and not the N-acetyl-D-glucosamine monomer, or highly crosslinked chitin[30]. It placed chitin at the plasma membrane of lipid-filled vesicles in the β-cells (Fig. 1c, e; Fig. 2) and as an intermediate layer in the secreted cement plaque. The intracellular location of the chitin signal would superficially exclude the possibility of it being chitin in the commonly accepted use of the term—i.e. a homopolymer of N-acetyl-D-glucosamine produced via a transmembrane chitin synthase. However, cellular production of chitin in crustaceans remains poorly understood and this arrangement may not be unusual. It has in fact been demonstrated that chitin synthase can be active in the membrane of specialized vesicles, secreting chitin into the vesicle lumen where it is accumulated ready for immediate release by exocytosis[36]. More intriguingly, Horst[37] provided evidence of lipid-bound intermediates during chitin synthesis by *Artemia salina*, which could provide insight into the co-location of lipid and chitin in the β-cell vesicles. While incompletely understood in arthropods, the interactions of chitin and lipids in chitosomes of fungi are well-described[38].

In this instance, the strongest evidence against the presence of chitin (sensu stricto)[29] bound to a chitin-binding protein, and supporting the likelihood of native chitin (a glycosylation), was the retention of CBD/WGA affinity through the SDS-PAGE procedure. Denaturing conditions would usually dissociate hydrogen-bonded chitin from chitin-binding proteins. Of course, this does not eliminate the possibility that native chitin, present as a protein glycosylation, could still bind to another yet unidentified chitin-binding protein during cement polymerization in a process akin to cuticular sclerotization. Although chitin and a Mr-lcp1-122k homologue were identified in the same locations on SDS-PAGE blots (Fig. 5a) and were identified in the same vesicles in the β-cells (Figs. 1c and 5b), chitin was concentrated at the vesicular membrane whereas the Mr-lcp1-122k homologue was found throughout the lumen of the vesicles. The faint signal from the latter may be a consequence of inter-species differences (68% identity of the Mr-lcp1-122k N-terminal sequence to the *B. amphitrite* homologue) or simply an indication of low abundance. Either way, we currently have no explanation for this inconsistency, other than to speculate that mixing could occur during secretion, or that chitin is, in fact, found throughout the vesicle lumen too, but concentrated at the periphery of the β-vesicles leading to the halo appearance in fluorescence microscopy (Figs. 1c and 2b).

The location of chitin in the secreted glue (Fig. 1i; red) appeared to be between the central proteinaceous core (green) of the cement plaque and the outer lipidic region. This spatial arrangement seemed counterintuitive due to the shared origin of lipid and chitin in the same β-cell vesicles and must have occurred via one of two secretion processes. First, α- and β-cell contents could simply have been released together followed by passive phase-separation, with the lipid-rich phase moving to the exterior and the protein/chitin mixture remaining towards the core. The thermodynamics underwater would presumably favour the opposite, however. In a scenario requiring more biological

control, the lipid and chitin mixture could alternatively be released first from the β-cells[20] with the proteinaceous contents from the α-cells released afterwards, physically forcing the lipid-rich material outwards with chitin retained at the protein/lipid interface by association with specific proteins. The consistent location of chitin in the secreted adhesive plaque implied a functional role. To confirm this, a series of experiments exposed attached cyprids to chitinase in order to compromise any involvement of chitin in adhesion. Almost all (98%) cyprids became detached after 8 h (Fig. 3) in the chitinase treatment and the action of the enzyme was shown to be directly on the adhesive plaque and not via degradation of the cyprid cuticle. Chitinase treatment had no effect, detectable by this method, on the adhesion of juvenile barnacles. This could be explained by the adhesive of the juvenile barnacle having less reliance on chitin, or that once 'cured' the adhesive was impervious to chitinase activity. Alternatively, it could simply be that access for the enzyme beneath the base of a juvenile barnacle was limited.

Having identified the presence of chitin and provided evidence for its functional role in adhesion, the objective was then to isolate cyprid cement proteins via the presence of chitin/native chitin and, thus, provide a basis for future work investigating the ancestral relationship between adhesion and cuticle formation in barnacles. Biotinylated WGA was used for this purpose. WGA is less specific than the CBD, binding between 3 and 4 β-(1,4)-linked N-acetyl-D-glucosamine residues[39]. In histological controls, it labelled some off-target locations in the musculature of the cement collecting system outside of the gland (Fig. 2) but, importantly, it also labelled the precise location of CBD-546 fluorescence in the β-cells (Fig. 2). Two WGA-reactive bands of equal intensity were present on a western blot of B. amphitrite cyprid cement gland proteins (Fig. 4), but a blot of whole cyprid extract presented equivalent bands of distinctly different intensity. As both samples contained 100 cement glands, the additional intensity of the upper band in the 50 whole cyprid sample suggested that the WGA-binding constituents of this upper band were not unique to the cement gland and were likely present elsewhere in the cyprid body. On the contrary, the lower band of the 100-gland western blot lane (Fig. 4) was of equal intensity in whole cyprid versus gland only samples and was therefore more likely to be cement gland-specific (note this is not obvious in Fig. 4 due to different exposure times used for presentation purposes). This was confirmed when the bands were subjected to analysis by MS/MS and mapped to a cement gland-specific library for M. rosa. Peptides in the upper band (Fig. 4; Supplementary Data 1) were absent from the M. rosa library and therefore presumably also present in the rest of the larval body. The lower band isolated from B. amphitrite contained a homologue to a previously unpublished cement gland-specific protein from M. rosa, namely Mr-lcp1-122k. This result was validated by immunohistochemical identification of the Mr-lcp1-122k homologue in the vesicles of the β-cells, as well as western blots with the same N-terminal specific antibody and the CBD.

To conclude, the data presented here demonstrate that chitin was produced in the β-cells of the cyprid cement glands and that it was released during adhesion to a surface. While stored in the β-cells, chitin surrounded the lipid and protein-filled lumen of secretory vesicles and, when released, associated itself with the protein-rich centre of the adhesive plaque. Chitin was essential for the proper functioning of the glue and, when enzymatically digested, the adhesive bond was substantially weakened. A cement gland-specific protein homologous to Mr-lcp1-122k was identified via the presence of chitin/native chitin. It currently seems most likely that native chitin is present on this protein as a glycosylation. Although poorly understood, such glycoproteins have been identified in recent studies of mineralization in

coralline algae[40] and molluscs[41]. Future work should therefore focus on the role of Mr-lcp1-122k in cyprid adhesive polymerization, as well as possible sequence or mechanistic similarity to proteins involved in cuticle formation and mineralization in barnacles and other taxa. Identification of chitin-binding proteins that may interact with the putative glycosylations of Mr-lcp1-122k may also yield new insight into the molecular mechanism of adhesive assembly during larval settlement. The observation that cyprid adhesion can be compromised so convincingly by the degradation of chitin could provide direction for novel, targeted fouling-control strategies. The widespread industrial interest in chitin, chitosan and their derivatives for all manner of industrial and biomedical applications[31] also raises the possibility that barnacles could take their place alongside mussels and tubeworms as inspiration for practical surgical adhesives.

## Methods

**Cyprid culture.** To obtain cyprids, adult barnacles (B. amphitrite) were removed from ASW (Tropic Marin, 33 ppt) for 18 h and then returned to fresh ASW, where they released stage 1 nauplius larvae. A point light source was used to attract the nauplii, which were collected by pipette. The settlement-stage cypris larva (cyprid) was reached in 4–5 days when nauplii were fed an ad libitum diet of Tetraselmis suecica at 28 °C. The cyprids were aged for 3 days at 6 °C prior to use in experiments[42].

**Tissue collection.** For collection of cement glands, cyprids were dissected[43] in pH 7.5 Trizma hydrochloride-buffered saline containing 0.1% Tween-20 (TBS-T), using sharpened tungsten needles (Fine Science Tools). For microscopy, glands were immediately fixed (4% formaldehyde freshly prepared from paraformaldehyde) for 1 h at room temperature (RT) then subjected to staining protocols. For analysis by western blot, 100 cement glands (from 50 individuals) were excised and any non-target tissue was removed before storage at −80 °C in minimal TBS-T. Glands were accumulated this way over a 5-week period from five different batches of larvae. All remained frozen prior to use, but it is important to note that freezing will have caused cell lysis and thus allowed mixing of the gland contents prior to preparation for SDS-PAGE.

**Confocal microscopy.** Fixed cement glands or secreted adhesive were either incubated directly with a solution of Nile Red (Sigma-Aldrich; 800 nM in phosphate-buffered saline (PBS) for 15 min), CellMask Deep Red plasma membrane stain (Thermo Fisher Scientific; 5 μg/mL in PBS for 30 min), or fluorescamine isothiocyanate (FITC, Sigma-Aldrich; 1.4 mM in PBS for 1 h) or, for immunohistochemistry, washed 6 × 10 min with TBS-T and blocked with 3% BSA in TBS-T overnight at 4 °C. After blocking, they were incubated with either chitin binding domain conjugated to Alexa Fluor 546 (CBD-546) at 1:20 dilution in PBS overnight at 4 °C, followed by 3 × 1 h washes[30], or 25 μg/mL of biotinylated WGA (WGA-biotin; Vector Laboratories) or 1:500 N-terminal Mr-lcp1-122k peptide antibody in blocking buffer for 1 h at RT, and then washed 6 × 10 min with TBS-T. WGA-labelled samples were incubated with a streptavidin-Texas Red conjugate (Vector Laboratories) diluted 1:400 in blocking buffer for 1 h at RT. Mr-lcp1-122k antibodies were incubated with 1:1000 goat anti-rabbit FITC conjugate (Sigma Aldrich) in blocking buffer. Prior to imaging, samples were washed 6 × 10 min with TBS-T and mounted in Prolong Gold mounting medium (Thermo Scientific). Imaging was conducted using a Leica SP8X inverted confocal microscope equipped with a pulsed white-light laser (470–670 nm) and two gated hybrid detectors with 40× (air) or 63× (oil) objective lenses. Additionally, imaging of individual vesicles was conducted on the same instrument using gated stimulated emission depletion (gSTED). All images were deconvoluted blind using 10 iterations and rendered as volume projections using the 3D module within Leica LASX.

**Protein preparation and electrophoresis.** Samples of cement glands, or whole cyprids, were homogenized in minimal 2× concentration NuPAGE LDS sample buffer (Invitrogen) on ice using a manual tissue grinder. In the case of the 100 cement glands, this was 30 μL. Samples were heated to 80 °C for 5 min, the appropriate quantity of 10× NuPAGE sample reducing agent (Invitrogen) was added and 25 μL of sample was placed in each well of a 1.5 mm 4–12% SDS-PAGE mini-gel (NuPAGE, Invitrogen). Gels were run for 60 min at 150 V. Colloidal blue staining was conducted following the protocol of Candiano et al.[44].

**Western blotting.** Proteins were electro-transferred onto PVDF membrane (Thermo Scientific) using Towbin's reagent and standard settings on a semi-dry Trans-Blot Turbo (Bio Rad). Ponceau staining was used to confirm successful protein transfer. The blots were washed 4 × 15 min with TBS-T at RT and blocked with 5% BSA in TBS-T overnight at 4 °C. For WGA affinity, blots were incubated for 1 h at RT with biotinylated WGA (Vector Laboratories) diluted in blocking

buffer at a concentration of 10 µg/mL. After 4 × 15 min washes with TBS-T, peroxidase-conjugated streptavidin (Vector Laboratories), diluted 1:20,000 in blocking buffer, was applied for 1 h at RT. Finally, membranes were washed 4 × 15 min in TBS-T at RT. WGA-conjugated bands were detected by chemiluminescence using Radiance Plus reagents (Azure Biosystems) on an Azure c280 gel documentation system with variable exposure times (10–30 s). To verify lectin specificity, 10 µg/mL of WGA was incubated in 200 mM hapten sugar (N-acetyl-D-glucosamine; Sigma-Aldrich) for 60 min and subsequently failed to detect the protein bands. For the antibody to the N-terminal peptide of Mr-lcp1-122k, the same protocol was followed, but an HRP-conjugated goat anti-rabbit antibody (Sigma-Aldrich) was used in the second stage. For the CBD, the process involved, first, application of a 1:2000 dilution of the SNAP-CBD fusion protein, followed by 1:2000 dilution of anti-SNAP (rabbit; New England Biolabs), followed by the goat anti-rabbit HRP-conjugate.

**Immunoprecipitation**. In preparation for mass spectrometry, 1000 whole cyprids were homogenized in 100 µL of TBS (pH 7.5) mechanically and by repeated freeze-thawing, followed by 4 °C centrifugation at 23,000×g for 15 min. Lysate was then made up to a final concentration of 150 mM NaCl, 1 mM EDTA, 5% glycerol and 1% NP40 containing protease inhibitors (PMSF 0.5 mM, leupeptin 10 µg/mL, aprotinin 10 µg/mL and pepstatin 10 µg/mL). This sample solution was frozen in aliquots at −80 °C until immunoprecipitation. Next, 10 µg of biotinylated WGA was incubated with the protein sample to form a complex. This complex was immobilized onto Pierce streptavidin magnetic beads (Thermo Scientific), which were then washed thoroughly following the manufacturer's guidelines. Proteins bound to the beads were eluted following magnetic precipitation using 2× Laemmli buffer and 90 °C heat, followed by SDS-PAGE and excision of the relevant bands. Bands were stored in 10 µL of 1% acetic acid prior to preparation for mass spectometry.

**Mass spectrometry**. Excised protein bands were destained by washing with ammonium bicarbonate and then 50% acetonitrile in ammonium bicarbonate. Gel pieces were subjected to reduction and alkylation with dithiothreitol and iodoacetamide, respectively. For digestion, sequencing-grade modified trypsin (Promega) was added at 1:50 w/w (enzyme: protein) and the reaction was incubated at 37 °C overnight. The reaction was stopped by addition of 1/10 volume 10% formic acid. The product was transferred to glass autosampler vials for LC/MS analysis. Peptide separation was performed on a Nanoflow Dionex 3500 RSLC system using C18 EASY-spray™ column (P/N ES803, C18 2 µm, 100 A, 75 µm × 50 cm; Thermo Scientific), maintained at 50 °C. Chromatographic separation was achieved with an acetonitrile gradient in water (2–80 %) over 100 min, using 0.1% w/v formic acid as the ion-pairing agent. The trapping column was a C18 Pep-Map™ 100, 5 µm 100 A (Dionex™) and was maintained at 45 °C. Data acquisition was performed in data-dependent analysis mode on a Q-exactive plus system (Thermo Scientific). Full scan MS$^1$ was performed at 70,000 MS resolution with an automatic gain control of 1e$^6$ and injection time of 100 ms. The scan mass range was 375 to 1400 m/z. The ten most abundant top ions were selected for MS/MS analysis with a normalized collision energy of 30. MS$^2$ data acquisition was performed at 17,500 MS resolution with an automatic gain control of 1e$^5$ and a maximum injection time of 100 ms. The isolation window was set to 1.3 m/z with an under fill ratio of 0.4%. Dynamic exclusion was set to 15 s and the full-width at half-maximum (FWHM) was 5 s.

**Mass spectrometry data analysis**. Raw MS/MS data were imported into Max-Quant (Max Planck Institute of Biochemistry) and, having selected the appropriate digestion protocol, searches were conducted using default settings with the minimum peptide length set to 7 amino acids. The initial library used for searches was a >200 nucleotide cut-off B. amphitrite transcriptome, representing all life-stages (ref. [32]), translated into amino acids in all six reading frames using EMBOSS Transeq (EMBL-EBI) on Ubuntu Linux v18.04 LTS. From the resulting spectral matches, protein groups (Supplementary Data 1) were selected for further analysis based upon the following criteria: matched peptides must be seven amino acids or longer, each predicted peptide group (putative protein) must contain matches to at least two contigs, and each matched contig must be identified by at least two unique peptides. These data were then compared to a refined cement gland-specific library (described below) for M. rosa using BLASTp (NIH) via Galaxy (galaxyproject.org). Contigs that constituted the B. amphitrite analogue of the M. rosa sequence were then also identified from the 6-frame translated B. amphitrite transcriptome using BLASTp (NIH) via Galaxy (galaxyproject.org). Alignment rates between the two species were calculated using BLASTp (NIH).

***M. rosa* cement gland-specific library production**. Briefly, ninety cement glands were isolated from mature M. rosa cyprids in cold 0Ca0Mg barnacle saline with 2.5 mM EDTA[11,30]. They were then transferred into 2× SDS-lysis buffer, treated with heat (98 °C for 5 min), and subjected to 10% SDS-PAGE. Major bands not present in a cyprid minus glands control were excised from SDS-PAGE gels and digested with trypsin. The digested peptides, fractionated by HPLC, were subjected to Edman degradation using a model 477A automated protein sequencer (Applied Biosystems, Inc.), connected on-line to a model 120A PTHAnalyzer (Perkin Elmer[45]). Separately, a full-length cement gland-specific secretory protein sequence database was

constructed by screening the M. rosa transcriptome, including ultra-deep sequencing of nauplius 6 and cyprid stages[46], using cement gland-specific expressed sequence tags, sequence data from a SMART library of the cement gland complex (cement glands with antennules), and Signal P search. This database was then screened using partial peptide sequences produced by Edman degradation of proteins isolated from the initial gel bands. Five sequences (Supplementary Table 2) were found in one protein with a predicted mass of around 122 kDa, which we refer to here as Mr-lcp1-122k (121670.63 Da, 1082 amino acids, pI = 11.67).

**Chitinase assay**. Chitinase from *Streptomyces griseus* was acquired as a lyophilized powder from Sigma-Aldrich. Working solutions were produced in filter-sterilized ASW (0.22 µm nitrocellulose; Millipore), adjusted to the enzyme's optimal pH of 6.0 to a target concentration of 1 mg/mL. Solubility was incomplete following extended vortex-mixing, so the solutions were spun down on a Thermo Scientific single-speed microcentirifuge for 1 min to remove pelleted residue prior to use of the supernatant in experiments. For the removal experiment, 100 3-day-old cyprids of B. amphitrite were allowed to settle on the base of 3-cm diameter polystyrene Petri dishes (Bibby Sterilin). After 18 h, unsettled individuals were removed by brief rinsing with distilled water and immediate re-immersion into ASW. The number of permanently attached cyprids and metamorphosed juvenile barnacles was counted. Three solutions were prepared: 1 mg/mL chitinase in pH 6.0 ASW, 1 mg/mL heat-treated chitinase (10 min at 65 °C) in pH 6.0 ASW and pH 6.0 ASW only. Six replicate dishes were used for each treatment. At the beginning of the experiment, 2 mL of the solution was added to each dish, and all dishes were placed on an orbital shaker set to 60 revolutions per minute at RT. Every hour, all dishes were removed from the orbital shaker and the number of attached individuals of each life-stage was enumerated. Detached individuals were evident floating in the medium.

**Reporting summary**. Further information on research design is available in the Nature Research Reporting Summary linked to this article.

## Data availability
The sequence information for Mr-lcp1-122k is freely available through NCBI accession number MK490677. Data produced by mass spectrometry and the chitinase assay are provided in Supplementary Data 1 and 2, respectively.

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

## Acknowledgements

The authors thank New England Biolabs for the supply of transfected *E. coli*. The research was supported by Office of Naval Research (ONR) grants N00014-16-1-2988 to ASC and N00014-16-1-3125 to A.S.C. and N.A., as well as ONR Global grant N62909-16-1-2215 to N.A. V.B.S.C. was supported by a Clemson University Research Foundation grant to A.S.M. K.O. was supported by JSPS-KAKENHI Grant Numbers JP12460087 and JP21651056, and the President Grant of Akita Prefectural University (S26-S28). K.O. also thanks Drs. Shogo Matsumoto, Atsushi Ohnishi and Naoshi Dohmae for peptide sequencing. N.A. wishes to thank the Newcastle University Bioimaging Unit for microscope time, Dr. Marcelo Rodrigues for helpful input during sample preparation, Prof. Gary Black (Northumbria University) for his assistance with mass spectrometry experiments, Prof. Timothy Ravasi (King Abdullah University of Science and Technology) for discussions regarding the *Balanus amphitrite* transcriptome, Prof. Dan Rittschof/Beatriz Orihuela (Duke University) for provision of barnacle broodstock and the Defence Science and Technology Laboratory (DSTL) for facilitating the collaboration with K.O.

## Author contributions

N.A. and V.B.S.C. designed and performed the experiments and wrote the paper. K.E. performed the mass spectrometry experiments. K.O. provided significant background data and expert input. A.S.C. and A.M. provided practical, financial and intellectual support to the research.

## Competing interests

The authors declare no competing interests.
