## [Peer Review File · Communications Biology]

Reviewers' comments:

Reviewer #1 (Remarks to the Author):

Refer to review report attached.

Reviewer #2 (Remarks to the Author):

N. Aldred et al. investigated the adhesive composition of the barnacle settlement-stage larva, the cyprid, and found first evidence for the presence of chitin within the vesicles of cement glands and the adhesive cement. Treatment of settled cyprids with chitinase induced their detachment over the following hours, which might indicate that chitin is essential for cyprid attachment at this stage. Next, the researches aimed to identify chitin-binding proteins and suggested a cement-gland-specific protein to form a chitin-protein complex within the adhesive. The authors further propose the intriguing idea that barnacle adhesion could have evolved from a modification of the cuticle secretion process.

Comments:

The study is addressing the question if chitin-protein complexes are involved in barnacle attachment. The manuscript is very well written and the presented results appear to be technical sound. However, while carefully reading the manuscript, several questions on the experimental approach and interpretation of the data arose that require further clarification.

Major concerns:

1. The treatment with chitinase induced detachment of cyprids larvae only prior to metamorphosis, juvenile barnacles remained attached. Are control animals and treated animals at the end of the experiment (8 hours) at the same stage? Are detached treated animals still alive? It seems likely that chitinase treatment is blocking the process of metamorphosis and thereby causes the animals to die, which might cause them to detach in a more indirect way. Please comment on this issue.
2. I admit that I do not understand the approach to identify the potential chitin-binding protein. The authors state that they "analyzed cyprid adhesive proteins that bind chitin by targeting their WGA affinity." I first assumed that the approach was indirectly: targeting chitin-bound proteins by using the chitin-binding WGA. However, it is described that denaturing conditions in the gels were used, which should separate non-covalent bindings. If chitin was bound to the candidate protein also under this conditions (which might be possible) the molecular weight observed in the western blot should be higher than the predicted weight of the protein alone.

Looking at the data, I suggest that the authors indeed identified a novel cement gland-specific glycoprotein with N-acetyl-D-glucosamine residues. It is stated in the discussion that this is "rather unlikely", as only a single, non-conserved, NxS moiety is present in the protein sequences. However, as the shown sequences in SUP3 are partial (also for *Megabalanus rosa*), this cannot be stated at this stage.

The authors convincingly show the identification of chitin in the cyprid adhesive, and the identification of a new, highly interesting cement gland-specific protein. However, from the presented data, there is no clear evidence that chitin and the novel identified protein Mr-lcp1-122k interact.

If I am mistaken and there is an indication that chitin-binding proteins should be directly recognized by WGA, please clearly explain this to avoid confusion of non-specialist readers.

Minor comments:

- Given that the production of the chitin-specific fluorescent probe is laborious and time-consuming, it is justified to use WGA for further experiments, especially as they both stain the same areas in vesicles and adhesive. However, it would be useful to clearly define their binding specificity early on in the manuscript. For example: the staining of antennules with WGA, but not with the chitin-binding fluorescent probe.
- Fig1: To place the letters (G-I) in the middle of the presented images is a bit confusing.
- The candidate protein is named Mr-lcp1-122k throughout the manuscript, but is called Lsp110k in the SUP 3.
- Was a chitin synthase found in the cement gland specific library?

Reviewer #3 (Remarks to the Author):

As the authors state, in this work, they go beyond proteomics analysis of marine adhesives and examine if other biomacromolecules play a role in the bioadhesion of barnacle larvae. While the identification of chitin at the cyprid adhesion site is unsurprisingly given the autofluorescence of this tissue in prior literature (Essocks-Burns et al., *J Exp Biol* (2017) 220, 194), the authors thoroughly investigate both the origin of the chitin and its contribution to the adhesive strength of the cement. Interestingly, the authors observe the presence of chitin has a profound impact on cyprid adhesion despite the predominance of phosphorylated proteins in the cement plaque that have been linked to adhesive strength in various marine adhesives. This work is well-written, and the conclusions are thought-provoking. I recommend that this manuscript be published with minor revisions (detailed below).

Pg 3, 2nd paragraph. The authors state WGA is more experimentally flexible and used in all subsequent experiments. The authors should replace "experimentally flexible" and elaborate on the benefits of using WGA instead of the recombinant CBD here rather than waiting until page 5 to explicitly state the limitations of the CBD.

Up until the discussion, the authors take care to describe chitin-protein complexes as an association between the two, which means the protein could be chitin-binding or glycosylated. In the discussion, they go back and forth with the terms association and interaction, the latter inferring the protein is chitin-binding. This should be more consistent for clarity. In particular, it is unclear in the last paragraph on page 6 if the authors are making a case for assigning the functionality of the identified protein one way or the other. In the last sentence on page 6, the authors suggest the protein contains little if any glycosylation. Does this mean the authors believe the protein is most likely chitin-binding rather than glycosylated? If so, the author should address why WGA binds to the protein even after heat treatment in SDS. If chitin has been shown to remain bound to other chitin-binding proteins under similar treatment it should be referenced.

Have the authors de-glycosylated a tissue collection prior to SDS-PAGE to see if either of the top two bands appreciably shift?

Dear reviewers,

The authors would like to express their gratitude to the three anonymous reviewers for carefully reading and considering our manuscript. From the comments it was clear that changes to certain areas of the manuscript could significantly improve its clarity, and that some further work would assist with this. We have therefore conducted additional experiments and revised the manuscript accordingly.

The original paper was composed of three results sections, each with a single significant finding. First, it was demonstrated that chitin was present in the cement gland β -cells and cement plaques of cyprids, using a chitin-specific molecular probe, the CBD. Second, it was shown that exposure of attached individuals to chitinase promoted detachment. Comments on these two findings were minor and are addressed below in the point-by-point response to reviewers.

The major question raised by two of the reviewers related specifically to the third experimental finding; the identification of a previously undescribed protein that is specific to the cement glands and that was identified by affinity to wheat germ agglutinin (WGA), which binds to chitin as well as short polymers of N-acetyl-D-glucosamine. The reviewers were convinced by the identification of this protein, but asked that we clarify and/or be more specific whether we believe this to be a chitin-binding protein *sensu stricto*, or a glycoprotein with N-acetyl-D-glucosamine residues. It is worth mentioning that this final point was left intentionally open in the original paper. Nevertheless, we have conducted substantial further work to try to resolve the question, including use of an antibody raised to an N-terminal peptide of the novel protein, as well as further experiments using the CBD. While we still do not have complete clarity on the chitin-protein relationship, we hope that the additional data included in this revised manuscript will interest the reviewers. It seems most likely that the chitin we have identified is present as a glycosylation, and we have added a note on terminology relating to this. We have also confirmed the location of the identified protein to be in the same vesicles as the chitin presented in the original manuscript, so strengthening our results. The manuscript has been extensively modified to address these general questions regarding the nature of the chitin/protein identified, so it has not been possible to outline all minor changes here. However, we have highlighted in yellow any major alterations to the original text of the main document in response to specific reviewer comments and provide line numbers in the responses below.

Point-by-point response to reviewers:

Reviewer #1 (Remarks to the Author):

The authors explored their hypothesis that the presence of chitin is an essential component of the barnacle cyprid adhesive. The paper systematically discussed the methods used to detect the locations of three components - lipids, protein and chitin in the β -cells of the cement gland and in the secreted cement plaques. A direct approach of using chitinase to probe the critical importance of chitin during the cyprids' settling phase was done with appropriate controls. Finally, much work was done to isolate cyprid cement gland proteins that associate with chitin, then subjected to LC-MS/MS peptide mass fingerprinting and a homologue to Mr-lcp1-122k was identified.

The findings and evidence of chitin being crucial in the adhesion process is novel as proteins have been thought to be the most important component of their adhesive. These discoveries will bring new insight and inspiration to the scientific communities from diverse fields – biology, marine science, anti-biofouling, materials science, biomedical etc., and might possibly influence how the assembly and structural information of the proteins can be studied, as well as how anti-biofouling surfaces/materials can be designed.

Abstract, introduction and conclusion were clear and concise. With regards to methods and statistical studies, methods were presented with sufficient details for experimental reproduction and chitinase assay was performed with appropriate population of cyprids and controls. Majority of the data presented are of high quality and good presentation, with some improvement required. References are ample, updated and relevant.

Overall the work presented is convincing, however there are a few points that the authors can help to answer and address in the paper if necessary:

1) A thoughtful discussion questioned the location of the chitin in the secreted cement, recognizing that the locations of the three secreted cement components were counterintuitive considering the location of chitin and lipids in the cells. Could the location of the lipids after secretion be due to natural minimization of entropy with simple hydrophilic – hydrophobic interactions, whereby the hydrophilic end of the lipids naturally prefer to be in contact with the seawater, while the hydrophobic tails face the inner chitin content? Thereafter the α -cells secrete proteins to the chitin-rich plaque core and chitin-protein cross-linking occur.

If the molecules were surfactants, with highly hydrophilic and hydrophobic ends, then it would be easy to envisage how the reviewer's suggestion could produce a structured monolayer. However the lipid layer of the cement plaque is several microns thick and it is difficult to envisage how this could arise via simple entropic rearrangement. The 'strata' are very clearly defined and, often, the covering of lipid is incomplete, all of which point more towards a secretory control mechanism rather than passive rearrangement. We have not gone into detail on this point in the manuscript without compelling evidence either way.

2) What happens to the external lipid layer after secretion under natural circumstances?

In the hours following metamorphosis into a juvenile barnacle, when the cyprid cement plaque is in its final location between the substratum and the basal membrane of the juvenile, the plaque retains the morphology that we have described previously – i.e. a proteinaceous core and a lipidic surface layer. This is presumably impacted by the onset of calcification (in *B. amphitrite*) and, in images of older juveniles (e.g. Essock-Burns et al. 2017, JEB), the lipid layer does appear to thin. However, we do not know for exactly how long the morphology persists.

3) It would be interesting to perform a systematic time-series (hourly maybe) study on how the chitinase affects (a) the secreted lipid layer and (b) the degradation of chitin over the exposed period. Is there any information on the lipid layer – does the lipid layer gradually decrease until the chitinase can access the chitin beneath or does chitinase penetrate through the lipid layer?

This experiment was attempted during the original experimental work. Unfortunately (and rather surprisingly...) the addition of chitinase to the barnacle medium significantly accelerated the growth of the bacterial population, making clear images of subtle changes to the cement plaque impossible to collect. Several approaches were attempted but, ultimately, the experiments with chitinase were becoming prohibitively expensive and this avenue was abandoned. It would be an informative experiment, however, *as per* our previous work on Alcalase (Aldred et al. 2008, Biofouling).

Minor revisions recommended:

1. Abstract – “Understandably, previous research has focused on the...” spelling mistake.

Corrected

2. Figure 1 C and E – superimposition will be helpful for readers to visualize the co-localization locations of chitin and lipids.

Having attempted this it is actually not very helpful – it results in a blur that is difficult to interpret. We feel that the similarity between Figures 1C & E is sufficiently clear for the message to be evident to most readers.

3. Figure 1 C and D, E and F – presentation of stained gland unclear in terms of direction or plane of view, a representative line profile can help.

A brightfield image of a cement gland in the same orientation as those in Figures 1C&E has been moved from Figure 2 to Figure 1C (inset) to address this question through illustration of the entire cement gland structure. The gland is also visible at the same scale in Figure 1B.

- 4) Figure 3 A – improve presentation such that all data points are visible. A bar-chart or a change of marker type should work.

Done

5. Page 6 – “On the contrary, the lower band was of equal intensity in the ‘whole cyprid’ versus ‘gland only’ samples...” Indicate which ‘whole cyprid’ sample was being referred to. 1000 cyprids?

This refers to the equivalent western blot lane of 100 glands. This has been clarified in the text on lines 337-340 of the discussion.

6. Supplementary information to be combined into a single PDF except for SUP 1.

Done

I recommend to accept this manuscript with minor revisions.

Reviewer #2 (Remarks to the Author):

N. Aldred et al. investigated the adhesive composition of the barnacle settlement-stage larva, the cyprid, and found first evidence for the presence of chitin within the vesicles of cement glands and the adhesive cement. Treatment of settled cyprids with chitinase induced their detachment over the following hours, which might indicate that chitin is essential for cyprid attachment at this stage. Next, the researches aimed to identify chitin-binding proteins and suggested a cement-gland-specific protein to form a chitin-protein complex within the adhesive. The authors further propose the intriguing idea that barnacle adhesion could have evolved from a modification of the cuticle secretion process.

Comments:

The study is addressing the question if chitin-protein complexes are involved in barnacle attachment. The manuscript is very well written and the presented results appear to be technical sound. However, while carefully reading the manuscript, several questions on the experimental approach and interpretation of the data arose that require further clarification.

Major concerns:

1. *The treatment with chitinase induced detachment of cyprids larvae only prior to metamorphosis, juvenile barnacles remained attached. Are control animals and treated animals at the end of the experiment (8 hours) at the same stage? Are detached treated animals still alive? It seems likely that chitinase treatment is blocking the process of metamorphosis and thereby causes the animals to die, which might cause them to detach in a more indirect way. Please comment on this issue.*

We can confirm with confidence that cyprids exposed to chitinase proceeded through metamorphosis successfully and all survived for the duration of the experiment. No doubt there would be an effect of the enzyme on the developing juvenile cuticle, but any effect was sub-lethal and did not obviously delay the process of metamorphosis. Survival of juveniles was not monitored beyond the conclusion of the assay, at which point all remained alive. A note to this effect has been added on line 164 and highlighted in the revised manuscript.

2. *I admit that I do not understand the approach to identify the potential chitin-binding protein. The authors state that they “analyzed cyprid adhesive proteins that bind chitin by targeting their WGA affinity.” I first assumed that the approach was indirectly: targeting chitin-bound proteins by using the chitin-binding WGA. However, it is described that denaturing conditions in the gels were used, which should separate non-covalent bindings. If chitin was bound to the candidate protein also under this conditions (which might be possible) the*

molecular weight observed in the western blot should be higher than the predicted weight of the protein alone.

Looking at the data, I suggest that the authors indeed identified a novel cement gland-specific glycoprotein with N-acetyl-D-glucosamine residues. It is stated in the discussion that this is “rather unlikely”, as only a single, non-conserved, NxS moiety is present in the protein sequences. However, as the shown sequences in SUP3 are partial (also for Megabalanus rosa), this cannot be stated at this stage.

The authors convincingly show the identification of chitin in the cyprid adhesive, and the identification of a new, highly interesting cement gland-specific protein. However, from the presented data, there is no clear evidence that chitin and the novel identified protein Mr-lcp1-122k interact.

If I am mistaken and there is an indication that chitin-binding proteins should be directly recognized by WGA, please clearly explain this to avoid confusion of non-specialist readers.

The reviewer is correct in all of their points and we apologise for the ambiguity of our original discussion. The manuscript was left intentionally open regarding the nature of the chitin-protein relationship due to the lack of compelling evidence either way, but we have now edited the manuscript to more clearly present our thoughts in the light of new data (Figure 5 and accompanying discussion). In particular, the reviewer should now find the discussion section to be more focussed. We summarise the additional data in the introductory paragraph to this response and relevant passages can be found highlighted on lines 73-94, 290-304 and 354-360 of the revised manuscript.

Minor comments:

- *Given that the production of the chitin-specific fluorescent probe is laborious and time-consuming, it is justified to use WGA for further experiments, especially as they both stain the same areas in vesicles and adhesive. However, it would be useful to clearly define their binding specificity early on in the manuscript. For example: the staining of antennules with WGA, but not with the chitin-binding fluorescent probe.*

We have added a sentence according to the reviewer’s suggestion on lines 139-141 of the revised manuscript.

- *Fig1: To place the letters (G-I) in the middle of the presented images is a bit confusing.*

Corrected

- *The candidate protein is named Mr-lcp1-122k throughout the manuscript, but is called Lsp110k in the SUP 3.*

Corrected

- *Was a chitin synthase found in the cement gland specific library?*

There was no chitin synthase significantly differentiated in the cement gland (i.e. no cement gland-specific chitin synthase was identified), but future work will include investigation and localization of chitin synthases that are present throughout the body of the cyprid to determine if any of these are present in specific locations of interest in the cement gland and involved in adhesion.

Reviewer #3 (Remarks to the Author):

As the authors state, in this work, they go beyond proteomics analysis of marine adhesives and examine if other biomacromolecules play a role in the bioadhesion of barnacle larvae. While the identification of chitin at the cyprid adhesion site is unsurprisingly given the autofluorescence of this tissue in prior literature (Essocks-Burns et al., J Exp Biol (2017) 220, 194), the authors thoroughly investigate both the origin of the chitin and its contribution to the adhesive strength of the cement. Interestingly, the authors observe the presence of chitin has a profound impact on cyprid adhesion despite the predominance of phosphorylated proteins in the cement

plaque that have been linked to adhesive strength in various marine adhesives. This work is well-written, and the conclusions are thought-provoking. I recommend that this manuscript be published with minor revisions (detailed below).

Pg 3, 2nd paragraph. The authors state WGA is more experimentally flexible and used in all subsequent experiments. The authors should replace "experimentally flexible" and elaborate on the benefits of using WGA instead of the recombinant CBD here rather than waiting until page 5 to explicitly state the limitations of the CBD.

We have addressed this comment through inclusion of additional detail on lines 125-134 and 139-141 of the revised manuscript.

Up until the discussion, the authors take care to describe chitin-protein complexes as an association between the two, which means the protein could be chitin-binding or glycosylated. In the discussion, they go back and forth with the terms association and interaction, the latter inferring the protein is chitin-binding. This should be more consistent for clarity. In particular, it is unclear in the last paragraph on page 6 if the authors are making a case for assigning the functionality of the identified protein one way or the other. In the last sentence on page 6, the authors suggest the protein contains little if any glycosylation. Does this mean the authors believe the protein is most likely chitin-binding rather than glycosylated? If so, the author should address why WGA binds to the protein even after heat treatment in SDS. If chitin has been shown to remain bound to other chitin-binding proteins under similar treatment it should be referenced.

We accept that the original submission was ambiguous on this point, as discussed above, and in response to the comments of reviewers 2 & 3 we have now significantly modified the manuscript in the light of reviewer comments, new data (Figure 5 and associated discussion) and a more thorough evaluation of historic literature. Major changes are highlighted throughout the manuscript and we are confident that the change in emphasis provides for a more compelling discussion. Although we remain uncertain, we now state our opinion that the chitin is more likely present as a glycosylation.

Have the authors de-glycosylated a tissue collection prior to SDS-PAGE to see if either of the top two bands appreciably shift?

This is a very good suggestion. This observation was not possible because we could not isolate the pure protein and could only recognise it from its location on an SDS PAGE gel (which would shift unpredictably if deglycosylated, and therefore become confused with other bands) and via its associated chitin (which could be removed by deglycosylation methods rendering it undetectable). Also, it seems likely, based on the aa sequence of the protein, that any glycosylation would be O-glycosylation and therefore very challenging to remove. In summary, we have not done this due to the inherent ambiguity of the result, but it is something we are keeping in mind for future work – particularly now that we have a working antibody and therefore means to identify the protein other than via its associated chitin.

REVIEWERS' COMMENTS:

Reviewer #1 (Remarks to the Author):

The authors have attempted sufficient additional experiments to resolve the questions raised regarding the nature of the protein - Chitin-binding protein or glycosylated form, and additional explanations and clarifications greatly improved the manuscript. Other issues raised were also answered and resolved.

Overall it is a good quality manuscript.

Reviewer #2 (Remarks to the Author):

The initial concerns regarding the identification of a potential chitin-binding protein have been addressed and the authors provide additional experiments to strengthen their statements. The manuscript has been thoroughly rewritten and improved both in clarity and readability. Although the protein-chitin relationship is not clearly resolved yet, the authors now clearly state and discuss that issue. Overall the study provides a highly valuable work in the field of bioadhesion and biofouling and will surely inspire follow-up studies characterizing the role of chitin-protein complexes in adhesives. I strongly recommend the publication of the revised version of the manuscript.

Reviewer #3 (Remarks to the Author):

The author have adequately addressed all concerns in the revised manuscript.